# Report on the Symposium “Molecular Mechanisms Involved in Neurodegeneration”

**DOI:** 10.3390/bs8010016

**Published:** 2018-01-18

**Authors:** Giselle Pentón-Rol, Majel Cervantes-Llanos

**Affiliations:** Center for Genetic Engineering and Biotechnology (CIGB), Ave. 31 e/158 y 190, Cubanacán, Playa, P.O. Box 6162, Havana 10600, Cuba; majel.cervantes@cigb.edu.cu

**Keywords:** neurodegenerative diseases, molecular mechanisms, Alzheimer’s disease, Parkinson disease, multiple sclerosis

## Abstract

The prevalence of neurodegenerative diseases is currently a major concern in public health because of the lack of neuroprotective and neuroregenerative drugs. The symposium on Molecular Mechanisms Involved in Neurodegeneration held in Varadero, Cuba, updated the participants on the basic mechanisms of neurodegeneration, on the different approaches for drug discovery, and on early research results on therapeutic approaches for the treatment of neurodegenerative diseases. Alzheimer’s disease and in silico research were covered by many of the presentations in the symposium, under the umbrella of the “State of the Art of Non-clinical Models for Neurodegenerative Diseases” International Congress, held from 20 to 24 June 2017. This paper summarizes the highlights of the symposium.

## 1. Introduction

Neurodegenerative diseases (NDs) are a heterogeneous group of disorders characterized by the progressive deterioration of the structure and function of the central nervous system or the peripheral nervous system. NDs affect millions of persons worldwide, and Alzheimer’s disease (AD) and Parkinson’s disease (PD) are the most common types [1].

NDs are characterized by a gradual loss of neurons and synaptic connections, usually occurring later in life and often leading to fatal outcomes. The specific diseases are distinguished by the presence of characteristic symptoms that depend on the location in the brain of the neuronal loss. Typically, the degree of neuronal loss correlates directly with the appearance and progression of clinical symptoms. In AD, neuronal loss occurs early in the hippocampus, a brain region involved in declarative episodic memory. In PD, the characteristic clinical signs of tremor, bradykinesia, and postural instability become evident only after 70–80% of the dopaminergic neurons in the substantia nigra are lost [2].

A variety of neurodegenerative disorders affect the brain or spinal cord of humans. The degeneration of nervous tissue can result from acute injury and chronic disease. An acute neuropathology in adults, children, and infants can arise from head or spinal cord traumas, infection, toxicity, liver failure, and cerebral ischemia resulting from stroke, cardiac arrest, or asphyxia. A chronic, progressive neuropathology occurs in adult disorders such as Alzheimer’s disease, amyotrophic lateral sclerosis (Lou Gehrig’s disease), Huntington’s disease, multiple sclerosis (MS), and Parkinson’s disease, and in infants and children with spinal muscular atrophy. These diseases involve the degeneration of neurons and glial cells [3].

Animal models and in vitro research become highly valuated, because the brain is almost unreachable. These models are used to study disease pathogenesis and also to ensure disease mimicry in at least a few accurate mechanisms, in relation to which new therapeutics may be tested.

The “State of the Art in Non-clinical Models for Neurodegenerative Diseases” International Congress was held in Varadero, Cuba, from 20 to 24 June 2017. This meeting offered an update on the molecular mechanisms and non-clinical models, from basic sciences to the proof of concept in clinical practice. The program covered a number of topics including the main NDs, and considered the pathways and molecules involved in the development of drugs.

National and international experts in NDs, both scientists and clinicians, shared and discussed recent advances in this fast-moving field. Clearly, the ultimate goal of all these efforts is to combat and eventually cure these disabling diseases affecting millions of individuals throughout the world. A better understanding of the molecular processes underlying NDs has led, in recent years, to the design of a good number of new therapeutic approaches, mainly taking into account the progressive nature of these diseases.

This report deals with interesting aspects treated separately in this International Congress in a symposium called: “Molecular mechanisms involved in neurodegeneration” (see Table 1).

The first session chaired by Professor George Perry and Professor Hartmut Wekerle opened the symposium.

## 2. Advances on the Pathogenesis and Drug Development for Multiple Sclerosis

Professor Wekerle’s conference entitled: “Infernal Trio—Neurodegeneration, Inflammation, and Gut Flora” focused on the influence of the microbiome in NDs, particularly multiple sclerosis (MS). This was in line with recent advances that facilitate the study of host immune response-shaping by the microbiota. Professor Wekerle exposed the relevance of inflammation for neurodegenerative diseases and introduced the term “gut flora”, a recently accepted participant in this and many other pathogenic processes, particularly, in those with an autoimmune background, as in the case of MS and the experimental disease model experimental autoimmune encephalomyelitis (EAE).

Results of the study in a rodent model of spontaneous EAE were presented, by which the researchers established that disease-triggering depended crucially on an intact gut flora [4]. Mechanisms linking the activation of brain-reactive T lymphocytes with gut-associated lymphatic tissues, and microbial components responsible for EAE pathogenesis with disease initiation were proposed. These findings will be further studied in MS patients [5].

Most of the approved therapies for MS are immunomodulatory, despite the complex pathogenic processes taking place and presenting signs of autoimmunity, inflammation, and demyelination or primary oligodendrocyte loss. The current knowledge of this disease demands for broader therapeutic strategies to obtain better efficacy.

MSc Majel Cervantes-Llanos tackled the subject of C-Phycocyanin (C-PC). C-PC is a phycobiliprotein found in large amounts in Spirulina platensis, a blue-green alga. Several reports document its pharmacological characteristics as a strong antioxidant with anti-inflammatory properties [6].

A reduction of disease scores and duration was observed by the administration of the C-PC in the EAE animal model induced with the MOG_35–55_ peptide in C56Bl/6 mice. The treatment with C-PC also increased regulatory T cell subset markers in the EAE model [7] and suggested that C-PC, because of its immunoregulatory effects as well as antioxidant and anti-inflammatory properties, targets the neurodegenerative component of the disease, with the probable additional benefit of its lower toxicity, derived from its natural origin. Moreover, results of the C-PC effect on proliferation of splenocytes from 2D2 transgenic mice, after MOG_35–55_ peptide stimulation, were shown. The presence of C-PC in the culture, prevented the splenocytes from developing an mRNA inflammatory pattern, known to be responsible for the migration of activated immune cells into the CNS.

On the other hand, Dr Pentón-Rol lectured on Phycocyanobilin (PCB), the tetrapyrrolic ring of the C-PC, and the lecture was focused on the demonstration of the remyelinating properties of PCB, as shown by the differential expression of remyelinating and demyelinating genes in the brains of animals treated with PCB in the EAE and stroke animal models [8].

The downregulation of genes linked to demyelination (LINGO1, Notch-1) and the upregulation of genes involved in remyelination (MAL, CXCL12) in the brain of EAE animals treated with PCB by the oral route, demonstrated PCB effect on the disease. Furthermore, in an ischemia reperfusion model induced by endothelin-1, the presence of colloidal gold immunolabeling was observed in oligodendrocytes in the white matter of the brain, evidencing the presence of CNPase (2′,3′-Cyclic-nucleotide 3′-phosphodiesterase) and neurofilaments. PCB preserved the mitochondria and the myelin sheath which was evidenced by an anti-MBP antibody. These results support the remyelinating effect of PCB and suggest its therapeutic potential in MS [9] and stroke.

The results presented by MSc Cervantes-Llanos and Dr. Penton-Rol support the therapeutic potentials of C-PC and PCB and their additional effects when combined with immunomodulators such as IFN beta [10].

## 3. Natural Products and Neurodegenerative Diseases 

The screening of compounds of natural origin with promising therapeutic properties that are relevant for the treatment of NDs was very well documented in the symposium.

Dr. Pamela Maher delivered a presentation entitled “Natural Products and their Derivatives for the Treatment of Age-Associated Neurological Disorders” [11,12].

The absence of drugs to halt the progression of any age-associated NDs is likely due to the failure of drug developers to recognize that there are mutations predisposing individuals to these diseases as they get older and to the fact that the vast majority of NDs arises from a combination of multiple toxic insults. Pamela expressed that it is unlikely that the current “single target” approach will yield useful drugs for these conditions. Taking into account the multifactorial nature of the neurodegenerative diseases in the elderly, the identification of multitarget-led-compounds is needed. While proposing a selection based on screening assays that reflect the biology of the aging brain, Dr. Maher anticipated that this approach to NDs drug discovery will likely produce safe and effective drugs.

Dr. Maher announced the completion of a series of proof-of-principle experiments demonstrating that some polyphenolic natural products were exceptionally effective in halting the progression of age-associated neurodegenerative diseases and ischemia in a wide variety of animal models. Furthermore, the potency and chemical medicinal properties of these compounds can be dramatically improved without losing their multitarget activities. The approach of single-target, high-affinity drugs to treat NDs is considered to be erroneous since it has failed to identify the compounds that halt NDs progression. Dr. Maher observed that drug discovery paradigms based upon phenotypic screening are more likely to succeed.

Researchers from the University of Tolima, Colombia, reported their early research using extracts of natural origin. They covered relevant lines of research like Parkinson, stem cells, and tumors.

Julian Guzmán-Varon presented evidence on the effective properties of *Mucunapruriens* (Mp) for the treatment of PD, consisting in reducing l-DOPA-induced dyskinesias in a dyskinesia model in hemi-parkinsonized Wistar rats [13]. Rats with dopaminergic injury induced with neurotoxin 6-hydroxydopamine and dyskinesia provoked by l-Dopa and benserazide applied for 19 consecutive days were treated with two dose levels of Mp intraperitoneally. Abnormal involuntary movements (axial, anterior limb, orolingual, and locomotor) improved with the Mp treatment and the lowest concentration was the most effective one.

Similarly, Garzón referred that the extracts of *Salvia miltiorrhiza* were found to have a positive effect on mouse mesenchymal stem cells, as demonstrated by a qualitative morphological evaluation of the neural phenotype. The author reported that the effects of *Salvia scutellariodes* depend on several metabolites, such as saponins, anthocyanins, phenols, and coumarins among others that could explain the properties of the plant extract [14].

Laura Lozano presented the results on the cytotoxic effect of the *Rhopalurus junceus* scorpion poison on the T98G cellular line derived from Glioblastoma (GB), the most common primary tumor of the CNS, which is practically untreatable. The poison of this scorpion has been used as an active ingredient in homeopathic medicine, improving the quality of life and survival of patients by relieving pain and symptoms. This researcher showed in vitro evidence of the cytotoxic effect at different doses and time points of this venom on the T98G cell line. The experimental endpoints were membrane integrity examined by staining with trypan blue or propidium iodide and cell viability evaluated using MTT and Apoptosis–Necrosis methods [15].

Discoveries in pharmacology have led to the evolution from the traditional “popular wisdom” to the successful development of drugs mainly based on a natural source. A more recent approach is the in silico research that enables the virtual screening of bioactive compounds and their molecular targets.

## 4. Alzheimer’s Disease: From Virtual Screening to Therapeutic Options

There were 26.6 million cases of AD in the world in 2006, and it is projected that AD worldwide prevalence will grow fourfold by the year 2050, with 106.8 million cases, i.e., 1 out of every 86 persons will be living with AD. This disease is the most prevalent form of dementia that predominantly affects the elderly. For this reason, in this [16] “State of the art” congress symposium, Alzheimer’s disease received a great deal of attention.

High-level domestic and international scientists addressed basic approaches such as the role of the mitochondria in oxidative stress, in silico evaluations of possible compounds with therapeutic potency, molecular mechanisms involved in brain plasticity, and therapeutic tools such as nanoliposomes for AD.

AD provokes extensive oxidative stress throughout the body, which is detected peripherally as well as in association with the vulnerable regions of the brain affected in the disease. As a disease of abnormal aging, AD demonstrates oxidative damage at levels that significantly surpass those of the elderly controls, which suggests the involvement of other additional factor(s). Since mitochondria are vulnerable to oxidative stress, it is likely that the interactions between mitochondrial dysfunction and oxidative stress contribute to the initiation and amplification of reactive oxygen species that are critical to the pathogenesis of AD.

Professor Perry showed evidence of the importance of oxidative processes on AD pathogenesis. The mitochondria undergo continual fission and fusion events, which regulate their morphology and distribution. A morphometric analysis showed a small but significant reduction in mitochondria numbers and their enlarged size in AD. The levels of the fission and fusion proteins DLP1, OPA1, Mfn1, and Mfn2C were significantly decreased in AD [17].

It has been demonstrated that the mitochondria were capable of fusing with each other but at a much slower rate in AβPP (amyloid-β protein precursor) overexpressing cells, and it was concluded that AβPP, through amyloid-β production, impaired the balance of mitochondrial fission and fusion through the regulation of the expression of mitochondria fission and fusion proteins.

Furthermore, Dr. Roberto Menendez talked about AMYLOVIS, a new family of compounds for the therapy of Alzheimer’s disease”. Dr. Menendez delivered an overview analysis of the results of AMYLOVIS [18].

AD is characterized by the presence of neuropathological brain deposits: senile plaques, formed by deposits of the β-amyloid protein (Aβ), and neurofibrillary tangles produced by tau protein hyperphosphorilation. At the Cuban Neurosciences Center a family of compounds called AMYLOVIS, have been obtained *in silico*, on the basis of their properties to inhibit the fibrillogenesis process. In vitro evaluations of the cytoprotective actions of these compounds were performed in cerebellar granule cell cultures exposed to cytotoxic stimuli, and their inhibitory effect on the aggregation of the amyloid beta peptide 1–42 was evaluated in cultures of humanized microglia cells.

The aggregation of the small β-amyloid (Aβ) peptide plays an important role in the development of AD, for which reason the understanding of the interactions of Aβ with aggregation inhibitors on an atomic level could be essential for the rational development of diagnostic and therapeutic tools.

Computational methods now play an integral role in modern drug discovery, and include the design and management of small-molecule libraries, the initial hit identification through virtual screening, the optimization of the affinity and selectivity of the hits, and the improvement of the physicochemical properties of the lead compounds [19].

In this symposium, the researcher Alberto Bencomo showed a study where virtual selections of parallel ligands were made and based on structures of 60 phenyl derivatives (PDC) with biological activity evaluated in vivo against amyloid. Using coupling techniques, the author described the possible interaction sites of PDC with Aβ. These compounds interacted preferentially with the amino acids S8–G9, E11–H13, Q15–L17, and F19, regions that are crucial for the formation of fibrils. Simulations showed that the hydrophobic and some polar interactions stabilize the formation of the ligand complex–amyloid. These studies suggested that these five PDCs could be used as potential inhibitors of amyloid aggregation and also in diagnostic assays [20].

In the same way, the researcher Samila Leon, showed an in silico study and the preliminary evaluation of a potential radiotracer of β-amyloid plaques present in Alzheimer’s disease. In silico evaluations and the preliminary assessment of a new naphthalene-derivative compound, ^18^F labeled, demonstrated that this compound is a novel potential tracer with a probable affinity to amyloid plaques. The parameters of this compound were in the range of values considered to be suitable for compounds capable of crossing the blood–brain barrier (BBB), and in fact ^18^F crossed the BBB in healthy mice and in the AD animal model. Also, its uptake and retention time were higher in transgenic mice compared to healthy mice [21].

Dr. Alvarez-Ginarte introduced tau proteins as promising targets for Alzheimer’s disease, since they are microtubule-associated and mainly found in the axon of mature neurons, and thier dysfunction is sufficient to cause neurodegeneration. Alzheimer’s disease, as well as a group of other neurodegenerative diseases known as tauopathies, is characterized by tau aggregation. Furthermore, the microtubule-binding region of tau, the hexapeptide motif (306VQIVYK311), can form intermolecular β-sheet structures and was presented as an excellent molecular target for new treatments of Alzheimer’s disease. Additionally, small compounds capable of disrupting the β-sheet structure of 306VQIVYK311, such as rhodanines, are reported as tau aggregation inhibitors. However, the hexapeptide interaction with this compound has not been fully elucidated. For this reason, Dr. Alvarez-Ginarte performed parallel docking and quantitative structure–activity relationship (QSAR) studies on 52 rhodanine derivatives, in relation with known in vitro IC_50_ values. The conclusion was that core structures can be positioned in the middle of the tau peptide backbone in these inhibitors. The rhodanine derivatives bind the peptide by embedding one aromatic group between the side chains of Ile308 and Tyr310, and the rest of the ligand interacts with the C-terminal half of the peptide (i.e., Val309, Tyr310, and Lys311). Furthermore, two-dimensional QSAR models were obtained using multiple linear regression analysis. So far, five compounds were identified as leaders, and a virtual screening strategy will be used to guide the identification of anti-Alzheimer lead compounds [22].

Expanding on more specific mechanisms based on omics data and their relationship with NDs, Dr. Fabrice Leclerc exposed a promising topic such as “Combined experimental and computational approaches in transcriptomics and interactomics of RNA-binding Proteins. Perspectives to decipher neurodegenerative disorders” [23].

Dr. Leclerc presented the concept of homeostatic balance in RNA–protein interaction networks that, when disrupted, leads to various human diseases associated with neurodegenerative disorders, as supported by the omics data.

Dr. Leclerc work used CPEB1 (Cytoplasmic Polyadenylation Element-Binding Protein 1) as a model system to analyze the transcriptome for the identification of genes regulated by changes in CPEB1 expression levels. The link between CPEB1 and FMRP (translational repressor fragile X mental retardation protein) was confirmed, and new partners were identified using 3D structures of well-known RBPs (RNA-binding protein). The binding specificity of RNA fragments in the perspective of application to CPEB1 and other targets was predicted.

The decline observed during aging involves multiple factors that influence several systems. This is the case of the learning and memory processes which are severely reduced with aging. It is admitted that these cognitive effects result from impaired neuronal plasticity, which is altered in normal aging and mainly in AD. Neurotrophins and their receptors, notably BDNF, are expressed in the brain areas having a high degree of plasticity and are considered to be genuine molecular mediators of functional and morphological synaptic plasticity [24].

Daymara Merceron-Martínez based her studies on previous results related to the fact that basolateral amygdala electrical stimulation produces a partial recovery of spatial memory in fimbria-fornix lesioned animals and it is also able to increase BDNF protein content in the hippocampus, a critical brain area for spatial memory recovery in these animals. Merceron-Martinez studied whether the increased BDNF protein content arises from previously synthesized RNA or from the de novo RNA expression. The de novo RNA synthesis in the hippocampus after amygdala electrical stimulation and training was measured. Data obtained from trained animals confirmed that the daily electrical stimulation of the amygdala during 15 min after water-maze training produced a partial memory recovery that was coupled to increased expression of BDNF and ARC genes in the hippocampus. Additionally, the acute study showed that a single session of amygdala stimulation induced a transient increase of both genes. These results confirmed the memory-improving effect of amygdala stimulation in fimbria-fornix lesioned animals. They also suggest that the memory-improving effect is mediated by newly synthetized BDNF acting on a memory relevant structure like the hippocampus. The increased amount of BDNF mRNA within the hippocampus seemed to be locally synthetized by mechanisms activated after amygdala stimulation [25].

Dr. Joseph H. Neale presented data on the precognitive effects of glutamate carboxypeptidase II (GCPII) inhibitors in animal models of schizophrenia, ethanol intoxication, and Alzheimer’s disease [26].

The extensive work of Dr. Neale is based on the fact that *N*-acetylaspartylglutamate (NAAG) is the third most prevalent transmitter in the nervous system of mammals [27]. After synaptic release, NAAG is inactivated by an extracellular enzyme, GCPII, and inhibitors of GCPII raise synaptic levels of NAAG and reduce the release of small amine transmitters through the activation by NAAG of type III metabolic glutamate presynaptic receptors. The results presented by Dr. Neale demonstrated that these inhibitors are effective in the animal models of several NDs, since they prolong the beneficial effects of the released NAAG [28,29].

Nanoliposomes as a therapeutic tool for Alzheimer’s disease, by Dr. Lara Ordóñez-Gutiérrez, completed the presentations on AD [30].

On the basis of the existence of a balance between Aβ in the brain and in the peripheral blood, Dr. Ordoñez-Gutierrez proposed the ‘sink effect’ by which a rebalancing towards a higher presence of Aβ in the blood would enhance peripheral clearance and thus reduce Aβ levels in the brain. Interestingly, the peripheral reduction of Aβ levels reduces the brain burden of the molecule. Therefore, and considering the limitations related to the crossing of the blood–brain barrier, nanoparticle derivatives were presented as therapeutic tools.

The study had two approaches: (1) the intraperitoneal injection of unilamellar nanoliposomes containing either phosphatidic acid or cardiolipin; (2) the use of a newly characterized monoclonal anti-Aβ antibody to construct immunoPEGliposomes with a high avidity for capturing Aβ in the periphery.

It was found that, firstly, liposomes containing phosphatidic acid or cardiolipin can alter the levels of the circulating amyloid peptide and directly or indirectly modify brain metabolism; secondly, that intraperitoneal injections of unilamellar vesicles containing the anti-Aβ antibody reduced the amount of Aβ 1–40 and 1–42 in both the plasma and the brain, and it was observed that this reduction was more important in the case of Aβ 1–42, which is of relevance for AD patients. Future studies are needed to clinically confirm the benefits of this approach [31].

## 5. Conclusions

The prevalence of neurodegenerative diseases is currently a major concern in public health because of the lack of neuroprotective and neuroregenerative drugs. In this symposium, we highlighted the molecular mechanisms and therapeutic approaches related to NDs, particularly AD.

The following messages were given: C-PC and PCB are promising therapeutic options for MS on the basis of their remyelinating effect in EAE models; the microbiome participates as a third player in MS pathogenesis; single-target approaches are insufficient for these multifactorial diseases, and molecules should be screened for different relevant target and effects; as often observed before, natural products continue to be an important source of drugs with beneficial effects for NDs; the roles of oxidative stress and mitochondrial performance are strongly linked to AD pathogenesis; in silico research is the leading approach in drug discovery and it is being widely used, mainly for AD.

The different methodologies and lines of research discussed in the symposium suggest new possible therapeutic options and provide further insights on NDs mechanisms, from basic to translational research, drug discovery (mainly from natural sources), and the use of high-technology methods., The hallmarks of the symposium were all focused on NDs. The congress provided a framework for translational research to advance towards new promising disease-modifying therapies for ND patients.

## Figures and Tables

**Table 1 behavsci-08-00016-t001:** Symposium’s short report and lectures summary.

Title	Author	Affiliation	Highlights
**Lectures**			
Role of mitochondria in the oxidative stress of Alzheimer’s disease.	George Perry, Ph.D.	Dean and Professor Semmes Foundation Distinguished University Chair in Neurobiology College of Sciences, The University of Texas at San Antonio, San Antonio, TX, USA	By affecting mitochondria fission and fusion proteins, β-amyloid damages mitochondrial normal functioning and contributes to oxidative stress process in AD.
Trio infernal—neurodegeneration, inflammation and gut flora.	Hartmut Wekerle, Ph.D.	Senior Professor, Neuroimmunology, Max Planck institute of neurobiology. Germany	Microbiome correlates with disease in animal models.
Combined experimental and computational approaches in transcriptomics and interactomics of rna-binding proteins. Perspectives to decipher neurodegenerative disorders.	Fabrice Leclerc, Ph.D.	Senior Scientist. Institute for Integrative Biology of the Cell. Paris, France	In silico design of specific RNA ligands (fragment-based approach) to restore the RBPs homeostasis in neurodegenerative disorders.
The peptide transmitter *N*-acetylaspartylglutamate: positive roles in animal models of schizophrenia, inflammatory pain, brain injury and cognition.	Joseph H. Neale, Ph.D.	Professor Emeritus. Department of Biology, Georgetown University. Washington, DC, USA	GCPII inhibitors are effective in animal models of several clinical and NDs since they prolong beneficial effects of released NAAG
Amylovis, a new family of compounds for the therapy of Alzheimer’s disease.	Roberto Menéndez, Ph.D.	Senior Researcher. Cuban Neuroscience Center. La Habana, Cuba	AMYLOVIS: family of compounds designed in-silico and their in vitro results on the cytotoxicity and the inhibitory effect on aggregation of the amyloid beta peptide 1–42.
Remyelinating effect of phycocyanobilin in animal models of multiple sclerosis and cerebral ischemia.	Giselle Penton-Rol, Ph.D.	Senior Professor, Center for Genetic Engineering & Biotechnology (CIGB). La Habana, Cuba	Phycocyanobilin (PCB) effect on differential expression of remyelinating/demyelinating genes and immunoidentification in the brain of animals treated with PCB in the EAE and stroke animal models.
Natural products and their derivatives for the treatment of age-associated neurological disorders.	Pamela Maher, Ph.D.	Senior Scientist. Salk Institute for Biological Studies, La Jolla, USA	Different in vitro models allow screening a high number of prospective drugs pharmacological properties.
Nanoliposomes as a therapeutic tool for Alzheimer’s disease.	Lara Ordóñez-Gutiérrez, Ph.D.	CIBERNED & Centro de Biología Molecular “Severo Ochoa” (CSIC-UAM) Madrid, Spain	Liposomes containing phosphatidic acid or cardiolipin can alter circulating amyloid peptide and intraperitoneal injections of uni-lamellar vesicles containing anti-Aβ antibody, reduced the amount of Aβ 1–40 and 1–42 in both plasma and the brain.
**Short Reports**			
Evaluation of the effect of s.s on the neuronal differentiation of mesenchymal stem cells isolated from mouse bone marrow.	Diana Garzón, B.S.	Research Group Experimental Models for Zoohumans Sciences. Faculty of Sciences. University of Tolima. Ibagué, Colombia	Preliminary results of the qualitative evaluation of the neural phenotype induced by *Salvia miltiorrhiza* on mouse mesenchymal stem cells.
*Mucunapruriens* as a phytodrug with antidyskinetic effect evaluated in model of dyskinesia in hemiparkinsonized wistar rats.	Guzmán V. Julian, B.S.	Research Group Experimental Models for Zoohumans Sciences. Faculty of Sciences. University of Tolima. Ibagué, Colombia	Pre-clinical evaluation of *Mucunapruriens* in a model of dyskinesia.
In silico study and preliminary evaluation of a potential radiotracer of β-amyloid plaques present in Alzheimer’s disease.	Samila León Ch. MSc.	Cuban Neuroscience Center, La Habana, Cuba	In silico evaluation and in vivo preliminary assessment of a new naphthalene-derivative compound, labeled with ^18^F as a novel potential tracer with affinity to amyloid plaques.
Amygdala electrical stimulation inducing spatial memory recovery produces an increase of hippocampal bdnf and arc gene expression.	Daymara Mercerón-Martínez, B.S.	Cuban Neuroscience Center. La Habana, Cuba	Basolateral amygdala electrical stimulation produces a partial recovery of spatial memory in fimbria-fornix lesioned animals increasing BDNF de-novo synthesis in the hippocampus.
Evaluation of the cytotoxic effect of *Rhopalurus junceus* poison on the t98g cellular line derived from glioblastoma.	Laura Lozano, B.S.	Research Group Experimental Models for Zoohumans Sciences. Faculty of Sciences. University of Tolima. Ibagué, Colombia	In vitro cytotoxic effect of the *Rhopalurus junceus* scorpion poison on T98G Glioblastoma cell line.
Docking and qsar studies of rhodanine derivatives as aggregation inhibitors for tau protein. Virtual screening strategy to guide the identification of anti-alzheimer lead compounds.	Yoanna María Álvarez-Ginarte, Ph.D.	Laboratory of Theoretical and Computational Chemistry, Faculty of Chemistry, University of Havana. La Habana, Cuba	Based on Tau aggregation and Rhodanines developed virtual screening strategy and identification of five compounds as leaders.
Use of qsar, docking and molecular dynamics techniques in the relational design of lead compounds for diagnosis and treatment of Alzheimer’s disease	Alberto M. Bencomo, MSc.	Cuban Neuroscience Center. La Habana, Cuba	Virtual selection of parallel ligands based on biologically active phenyl derivatives delivered five potential inhibitors of amyloid aggregation and also in diagnostic.
“In vitro” immunoregulatory mechanisms of C-phycocyanin.	Majel Cervantes-Llanos, MSc.	Center for Genetic Engineering & Biotechnology. La Habana, Cuba	Pre-clinical study of the oral effect of C-PC on de EAE model showing decrease of disease scores and duration and in vitro immunoregulatory and anti-inflammatory mechanisms.

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
