# Peer review of "Report on the Symposium “Molecular Mechanisms Involved in Neurodegeneration”"

_behavsci, 2018, doi:10.3390/bs8010016_

Round 1

Reviewer 1 Report

As a perspective / conference report, the paper can have a chance, however, the contribution of this paper would be limited, given the conference has abstract book, and some presenters will publish their work, if not already.

The conclusion is so broad, and non-specific, as it could be given before all this paper is written. Make the conclusion specific to this conference.

The paper currently has 4 citations. Each single argument that refers to scientific presentations should be supported by a conference presentation citation. All of them would be the conference, but each one would have a specific presenter (author), presentation title , and date.   Please revise accordingly.

The paper currently has several single sentence paragraphs. This should be minimized. Please try to have about 3 sentences for  your short paragraphs.

A part of this review highlights the lack of medications that are available. Such not could be a part of the conclusion.

Please add new citations of the field (outside the conference) to back the arguments made.

Author Response

Journal Behavioral Sciences (ISSN 2076-328X)

Manuscript ID: behavsci-252579

Type: Conference Report

Number of Pages: 15

Title: Report on the Symposium: Molecular Mechanisms Involved in

Neurodegeneration

Authors: Giselle Pentón-Rol * and Majel Cervantes-Llanos

Authors would like to thank for the revision and accurate comments and suggestions.  All of them were revised and the adequate corrections introduced.

First I would like to draw attention to the inclusion of Majel Cervantes-Llanos as author of this work due to her collaboration in the organization of the symposium and to the revision and corrections made to the manuscript.

We are submitting the revised article for your consideration. Due to the nature of the corrections suggested, this is a new version of the article and it was not possible to enumerate or specify the corrections made.

Comments and Suggestions for Authors

Reviewer #1

1.       As a perspective / conference report, the paper can have a chance, however, the contribution of this paper would be limited, given the conference has abstract book, and some presenters will publish their work, if not already.

As the reviewer commented, the report intends to highlight the work from researchers that have or will publish their work but also the data reported from interesting preliminary research. Also to point out those particular lines of work to which, researchers dedicate a great deal of attention. All having in mind that researcher can communicate different messages based on existing data analyzed and interpreted form a different perspective and experience.

2.       The conclusion is so broad, and non-specific, as it could be given before all this paper is written. Make the conclusion specific to this conference.

Authors agree with the suggestion made by the reviewer and added information to adequate the conclusions to the symposium.

3.       The paper currently has 4 citations. Each single argument that refers to scientific presentations should be supported by a conference presentation citation. All of them would be the conference, but each one would have a specific presenter (author), presentation title, and date.   Please revise accordingly.

Authors agree with the suggestion made by the reviewer. Conference presentation citations were added.

4.       The paper currently has several single sentence paragraphs. This should be minimized. Please try to have about 3 sentences for your short paragraphs.

Authors agree with the suggestion made by the reviewer and reduced the use single sentence paragraphs.

5.       A part of this review highlights the lack of medications that are available. Such not could be a part of the conclusion.

Authors agree with the suggestion made by the reviewer and made the corresponding correction in the conclusions.

6.       Please add new citations of the field (outside the conference) to back the arguments made.

Authors agree with the suggestion made by the reviewer. Citations to the arguments and when possible to author´s published work were added.

Reviewer 2 Report

The paper seeks to summarize and detail the proceedings from an international conference on mechanisms in neurodegenerative diseases. The content of the conference and the paper is appropriate for the journal and report details novel work in the area. The primary concern with the report, as written, is the style used to describe the conference proceedings. The report requires considerable revision across all sections to bring it in line with appropriate scientific reporting. The major issues are detailed below. 

1. Use of tense. The writing style changes from active to passive to present to past tense.  The begins in the abstract and is present on almost every page of the report.  It is suggested that, since the report is describing the proceedings of an event that has already taken place, the authors use past-tense throughout.

2. Use of informal/colloquial style (A). Each speaker is described separately, and in some cases, the accomplishments, title, and (subjective) career achievements of the speaker are included. Avoid subjective phasing such as "delivered an interesting and exciting presentation".  This is an opinion.  Please keep comments to an objective description, as per scientific/academic protocol.  *Note: It is suggested that speakers are introduced by name, title of presentation, and university (or organization) in an introductory paragraph -- or a table could be created that provides all these details. 

3. Use of informal/colloquial style (B).  The report jumps between first person, second person, and third person. Scientific writing should remain in third person. When describing the person's work, the description can be (for example) Dr. Wekerle or Professor  Wekerle. Please change all instances of my, his, her, etc; and please use last names (surnames) only.

4. Use of direct quotes from the speakers rather than summary description and paraphrasing. This is the most troublesome aspect of the report as currently written. Almost every page includes a paragraph long direct quote of a speaker's presentation. This diminishes the report to a transcript of the proceedings, which is not appropriate for a journal publication. Please either summarize these quotes into the main points (please DO NOT accidentally plagiarize), or ask each of the speakers to provide their own summary of their talk that you can place in the manuscript. This should follow the format of a scientific journal style. 

5. Please remove all descriptions of the impact of the presentation on the audience, or what may have followed the presentation; for example, Professor Perry's lecture... opened up an important debate on the basic aspects of AD."  Please review the report carefully for all instances like this.

6. To list the chronical order the presentations, please do not do so as an introduction to a section or paragraph. Instead, include the order in the introductory paragraph or Table (see issue #2).

Author Response

Journal Behavioral Sciences (ISSN 2076-328X)

Manuscript ID: behavsci-252579

Type: Conference Report

Number of Pages: 15

Title: Report on the Symposium: Molecular Mechanisms Involved in

Neurodegeneration

Authors: Giselle Pentón-Rol * and Majel Cervantes-Llanos

Authors would like to thank for the revision and accurate comments and suggestions.  All of them were revised and the adequate corrections introduced.

First I would like to draw attention to the inclusion of Majel Cervantes-Llanos as author of this work due to her collaboration in the organization of the symposium and to the revision and corrections made to the manuscript.

We are submitting the revised article for your consideration. Due to the nature of the corrections suggested, this is a new version of the article and it was not possible to enumerate or specify the corrections made.

Reviewer #2

Comments and Suggestions for Authors

The paper seeks to summarize and detail the proceedings from an international conference on mechanisms in neurodegenerative diseases. The content of the conference and the paper is appropriate for the journal and report details novel work in the area. The primary concern with the report, as written, is the style used to describe the conference proceedings. The report requires considerable revision across all sections to bring it in line with appropriate scientific reporting. The major issues are detailed below.

1.       Use of tense. The writing style changes from active to passive to present to past tense.  The begins in the abstract and is present on almost every page of the report.  It is suggested that, since the report is describing the proceedings of an event that has already taken place, the authors use past-tense throughout.

2.       2. Use of informal/colloquial style (A). Each speaker is described separately, and in some cases, the accomplishments, title, and (subjective) career achievements of the speaker are included. Avoid subjective phasing such as "delivered an interesting and exciting presentation".  This is an opinion.  Please keep comments to an objective description, as per scientific/academic protocol.  *Note: It is suggested that speakers are introduced by name, title of presentation, and university (or organization) in an introductory paragraph -- or a table could be created that provides all these details.

3.       Use of informal/colloquial style (B).  The report jumps between first person, second person, and third person. Scientific writing should remain in third person. When describing the person's work, the description can be (for example) Dr. Wekerle or Professor  Wekerle. Please change all instances of my, his, her, etc; and please use last names (surnames) only.

4.       4. Use of direct quotes from the speakers rather than summary description and paraphrasing. This is the most troublesome aspect of the report as currently written. Almost every page includes a paragraph long direct quote of a speaker's presentation. This diminishes the report to a transcript of the proceedings, which is not appropriate for a journal publication. Please either summarize these quotes into the main points (please DO NOT accidentally plagiarize), or ask each of the speakers to provide their own summary of their talk that you can place in the manuscript. This should follow the format of a scientific journal style.

5.       5. Please remove all descriptions of the impact of the presentation on the audience, or what may have followed the presentation; for example, Professor Perry's lecture... opened up an important debate on the basic aspects of AD."  Please review the report carefully for all instances like this.

6.       6. To list the chronical order the presentations, please do not do so as an introduction to a section or paragraph. Instead, include the order in the introductory paragraph or Table (see issue #2).

Authors agree with all the suggestions made by the reviewer and have introduced changes in the article to fulfill all recommendations.

Round 2

Reviewer 2 Report

Excellent attention to the revisions.  The report is comprehensive and presented appropriately.  This should be of interest to the readership, especially those interested in neurological diseases. There are a few remaining typos scattered across the paper. It is suggested that the authors make once last very careful proof-read and edit for the final draft (Note: several grammar-/typo- checking programs exit, such as Grammarly, which may speed the task).